# Population Genetic Structure of the Wild Boar (*Sus scrofa*) in the Carpathian Basin

**DOI:** 10.3390/genes11101194

**Published:** 2020-10-14

**Authors:** Bendegúz Mihalik, Krisztián Frank, Putri Kusuma Astuti, Dániel Szemethy, László Szendrei, László Szemethy, Szilvia Kusza, Viktor Stéger

**Affiliations:** 1Institute of Animal Science, Biotechnology and Nature Conservation, University of Debrecen, 4032 Debrecen, Hungary; mihalik.bendeguz@gmail.com (B.M.); krisz.frank.biol@gmail.com (K.F.); kusumastuti@live.com (P.K.A.); 2NARIC Agricultural Biotechnological Institute, 2100 Gödöllő, Hungary; szemethy.daniel@gmail.com; 3Doctoral School of Animal Science, University of Debrecen, 4032 Debrecen, Hungary; 4Department of Nature Conservation Zoology and Game Management, University of Debrecen, 4032 Debrecen, Hungary; szendrei@agr.unideb.hu; 5Faculty of Regional Development, University of Pécs, 7100 Szekszárd, Hungary; laszlo.szemethy@gmail.com; 6Animal Genetics Laboratory, University of Debrecen, 4032 Debrecen, Hungary

**Keywords:** Carpathian Basin, genetic diversity, microsatellites, *Sus scrofa*, wild boar

## Abstract

In the Carpathian Basin the wild boar (*Sus scrofa*) belongs among the most important game species both ecologically and economically, therefore knowing more about the basics of the genetics of the species is a key factor for accurate and sustainable management of its population. The aim of this study was to estimate the genetic diversity and to elucidate the genetic structure and location of wild boar populations in the Carpathian Basin. A total of 486 samples were collected and genotyped using 13 STR markers. The number of alleles varied between 4 and 14, at 9 of the 13 loci the observed heterozygosity was significantly different (*p* < 0.05) from the expected value, showing remarkable introgression in the population. The population was separated into two groups, with an F_st_ value of 0.03, suggesting the presence of two subpopulations. The first group included 147 individuals from the north-eastern part of Hungary, whereas the second group included 339 samples collected west and south of the first group. The two subpopulations’ genetic indices are roughly similar. The lack of physical barriers between the two groups indicates that the genetic difference is most likely caused by the high reproduction rate and large home range of the wild boars, or by some genetic traces’ having been preserved from both the last ice age and the period before the Hungarian water regulation.

## 1. Introduction

The wild boar (*Sus scrofa*) is one of the most common big game species in Europe and worldwide. The origin of the species lives in East Asia, where the wild boar was separated from its closest relatives (*Sus verrucosus*) some 0.9–0.5 million years ago. Under the influence of the last glacial a strong decrease in numbers happened, but the Carpathian Mountains functioned as a refugia, thus many species, including wild boars found an area to survive here. When the ice age ended, these species rapidly recolonized the neighbouring areas and nowadays wild boars are widely distributed across Europe, Asia and Northern Africa [1,2,3,4,5,6]. Their population size has rapidly increased since the 1960′s: in Europe the 5-year growth rate varied between 1.40 and 1.73 from 1990 to 2010. In 2018 the estimated number of individuals in Hungary was 95850 [7]. At the same time, it is one of the world’s 100 worst invasive species, with a growing presence in urban areas [8,9,10]. The annual boar meat production is more than 6000 tons, and provides almost half of the whole game meat production in Hungary [7].

As a widely distributed species, wild boars have many subspecies: 16 of them were described in Europe alone [11,12]. Genetic diversity and genetic structure represent key components of the species’ survival, as large and diverse populations cope better with natural changes [13,14]. Different methodologies were used to study the genetics of wild boars. A well-established method is the analysis of mitochondrial DNA (mtDNA) sequences, especially the control region, also called D-loop, to identify phylogenetic patterns. This part of the mtDNA has a relatively low mutation rate, therefore changes appear on a larger time scale. Previous phylogenetic studies identified 94 haplotypes across Europe [15]. In the middle-eastern part of Europe—including the Carpathian Basin—a total of 16 haplotypes were found, forming two clades. The E1 haplogroup is common in Europe, whereas E2 seems to be endemic to the Italian peninsula and Sardinia. A few percent of Asian haplotypes from both the near-eastern and far-eastern clades were also found [3,15,16,17,18]. Also, 14 subpopulations were found in a study that covered most of Europe (excluding Hungary, Ukraine and Austria in Middle-Europe) [14].

A major drawback of mtDNA is its incapability of separating closely related individuals [17,19]. Microsatellite (STR) markers have been extensively used as molecular markers for high-resolution population assessments. Their multi-allelic nature, high levels of allelic richness, co-dominant mode of inheritance, and their easy PCR amplification and detection make microsatellites one of the most informative molecular markers for a relatively low price. By using multiple markers in one reaction, i.e., multiplexing, amplification can be made more time—and cost-effective. The development of novel microsatellite markers is cost and labour-intensive. However, due to cross-species amplification, microsatellite markers can be adopted to use in closely related species [19,20]. In the case of wild boars this is also an important consideration, because they can easily inbreed with domestic pigs and even with other suids, although this problem affects wild boar populations at a relatively low level [16,21,22].

In this study the main objective was to investigate the genetic diversity and genetic structure of wild boar populations/subpopulations across the region. The Carpathian Basin represents a crossroads of postglacial colonization routes and is a genetic hotspot for many terrestrial species [4,5,6]. However, the genetic or population structure of species in this region is not well documented. Genetic diversity and genetic structure bear signatures caused by gene flow, and by physical or biological barriers between populations. The identification of biological populations and subpopulations is relevant for population monitoring, culling plans and disease control, which could be applied to biological rather than administrative units. The population genetic structure and diversity of wild boars detailed here provides unique information for the development of management strategies aimed to maintain the highest possible level of genetic diversity.

## 2. Materials and Methods

A total of 486 samples were collected from wild boars shot at hunting sites by licensed hunters. Samples originated from free-ranging wild boar were legally shot during organized hunting events. For this no specific approval was needed, all applicable international, national and institutional guidelines were followed. This study was carried out in strict accordance with the recommendations and rules in the Hungarian Code of Practice for the Care and Use of Animals for Scientific Purposes. The protocol was approved by the Animal Care and Ethics Committee of the Agricultural Biotechnology Institute, NAIK. Hair follicles (*n* = 63) and tissue samples (*n* = 423) were collected in individually marked collection tubes. Muscle tissue samples were preserved in 96% ethanol, and all samples were stored at −20 °C until processing. For each sample, collection date and location were noted. The GPS coordinates of the kill sites were used if possible. In case if the exact location was unknown, the GPS coordinates of the centre of the hunting range were assigned to the sample in order to reach the highest possible accuracy. A few samples were also collected from neighbouring countries, namely Romania (*n* = 5), Croatia (*n* = 4), and Slovakia (*n* = 4) for comparison with the Hungarian samples.

Total genomic DNA was isolated from tissue samples using the Genomic DNA Mini Kit (Geneaid, New Taipei, Taiwan) and from hair follicles using the QIAamp DNA Investigator Kit (Qiagen, Hilden, Germany). The manufacturer’s instructions were followed in both cases. The quantity and purity of isolated DNA were checked by spectrophotometry in a NanoDrop ND-1000 machine (Thermo Fisher Scientific, Waltham, MA, USA). The DNA was stored at −20 °C until PCR amplification.

PCR amplification was performed using a multiplexed microsatellite marker set with 13 markers described by Lin et al. [23]. Amplification was carried out in a 20 µL reaction volume containing 10 µL of Multiplex PCR mix (Qiagen, Hilden, Germany), 7 µL of primer mix (10 μM) and 3 µL of template DNA (15–30 ng/µL). The PCR conditions were as follows: initial denaturation at 95 °C for 15 min, followed by 35 cycles of denaturation at 94 °C for 30 s, annealing at 61 °C for 30 s and at 72 °C for 60 s and a final extension at 72 °C for 90 min. Fluorescently labeled PCR products were separated on an ABI 3100 Genetic Analyzer (Applied Biosystems, Foster City, CA, USA), using LIZ 500 (Applied Biosystems, Foster City, CA, USA) as internal standard. Allele sizes were scored with Peak Scanner v1.0 software (Applied Biosystems, Foster City, CA, USA), and stored as individual genotypes in a Microsoft Excel table. To determine the number of populations and subpopulations and the genetic difference between them (Nei’s F_st_ value) the Geneland v.3.4.2. [24] program was used. The results were visualized by the QGIS 2.18.23 software based on Google Maps. The number of alleles per locus, the expected and observed heterozygosity values, deviations from the Hardy-Weinberg equilibrium after Bonferroni correction, and measures of genetic diversity for each locus and averaged across loci were calculated using the GenAlEx v.6.5. software [25]. To determine the possible barriers, the dataset was ran in the Barrier v2.2 software [26],

## 3. Results

### 3.1. Genetic Indices

A total of 291 male and 195 female wild boars were genotyped, and the STR markers used showed fairly high polymorphisms (Table 1).

The number of alleles varied between 4 (PigSTR14A) and 14 (PigSTR11B), with an average of 7.62. The effective number of alleles was between 1.162 and 5.095 (PigSTR14B and PigSTR15A, respectively), with an average of 2.29. Significant deviation from the Hardy-Weinberg equilibrium with He > Ho occurred in eight cases with *p* < 0.001 (PigSTR7B, PigSTR4C, PigSTR11B, PigSTR1B, PigSTR15A, PigSTR5C, PigSTR13E, PigSTR1A). A single He < Ho with *p* < 0.05 was also found (PigSTR14B). These results suggest strong introgression in the population.

### 3.2. Population Structure

Geneland detected a genetic structure and clustered the samples into two groups (with 80% probability). The first group included 147 samples from the eastern part of Hungary, whereas the second, larger one consisted of 339 individuals from the western part of the country (Figure 1). The F_st_ value was 0.02975, suggesting that gene flow between the groups is high enough for the two groups to belong into one population, but there are genetic differences between them. Most likely this means that there are two different subpopulations present in the region.

The subpopulations were almost perfectly separated geographically, but no physical barriers seemed to be present between them. There was one sample in the north-western region that clearly did not fit into the surrounding cluster, and there were four samples in the south-western region that differed from the surrounding samples. In the next step the output of the Barrier software was put on the map, but the results are not consistent with either genetic segregation or topography or the major road network of the area (Figure 2).

One of the greatest changes in the environment of Hungary was water management in the late 1800′s, therefore the previous results were put on a hydrographic map from the mid-1800’s (Figure 3).

No clear correlation can be detected in this case either; however, the location of the smaller subpopulation is roughly the same as the floodplain in the north-eastern part of the country bordered by the Mátra mountains.

### 3.3. Genetic Indices of Subpopulations

Per population diversity values were similarly high as that for the whole sample set (Table 2).

The number of alleles varied between 3–12 (PigSTR4B & PigSTR14A and PigSTR15A) and 3–13 (PigSTR14A & PigSTR1A and PigSTR11B & PigSTR15A) with an average of 5.62 and 6.54, respectively. The effective number of alleles was between 1.10 and 4.84 (PigSTR14B and PigSTR15A) and between 1.19 and 5.15 (PigSTR14B and PigSTR15A). In group 1 five loci showed deviation from the Hardy-Weinberg equilibrium; in four cases He > Ho with *p* < 0.001 significance level (PigSTR7B, PigSTR1B, PigSTR5C, PigSTR13E) and in one case He < Ho with *p* < 0.05 significance level (PigSTR11A) were found. In group 2, significant deviation from the equilibrium with He > Ho occured in eight cases with *p* < 0.001 (PigSTR7B, PigSTR4C, PigSTR11B, PigSTR1B, PigSTR15A, PigSTR5C, PigSTR13E, PigSTR1A), and with He < Ho in a single case with *p* < 0.05 level of significance (PigSTR14B).

## 4. Discussion

Our extensive sampling and the higher number of STR markers used resulted in 486 individual multilocus genotypes of wild boars across the Carpathian Basin. The genetic diversity revealed corresponds very well to the results of other studies shown below, although the markers used, and sample numbers differ greatly between studies. Frantz et al. [27] found an average allele number of 8.8 using 14 STR markers. Similar results were reported by Vernesi et al. [28] using nine markers in 29 Hungarian wild boars. In another study including 49 Hungarian wild boars, the average number of alleles across 14 STR markers was 6.21 [29]. The average allele numbers determined in Bulgaria and Germany based on 10 markers were 12 and 7.5, respectively [30]. In Croatian wild boars tested with 14 microsatellites the allele numbers ranged from four to 19 with an average of 8.92 [31]. Higher average allele numbers were found by Ferreira et al. [32] in 110 wild boars from Portugal: based on six markers, the allele numbers varied between three and 15, averaging 10.17. According to Velickovic et al. [3] 9–29 alleles were found with an average of 19 alleles per locus, in a fairly large sample set from 13 countries all across Europe, which should be the reason for high diversity. In a study that included samples from 10 regions all across East-Asia Choi et al. [33] found allelic diversities between 3.4 and 9.6 (average: 6.46) using 16 microsatellite markers on a total of 238 wild boars (Table 3).

The heterozygosity values found in our work are similar to the results of previous European studies but differ slightly from the results of Hungarian samples. The Bulgarian wild boars’ observed heterozygosity values were lower than expected in nine markers, and in Germany at eight loci [30]. Sprem et al. [31] also found lower observed heterozigosities than expected in 13 out of 15 groups, although the inbreeding coefficients were extremely low (0.004–0.172). On the other hand, Hungarian boars tested by Vernesi et al. [28] were almost completely in equilibrium, with only one marker deviating significantly. In Costa’s study there were eight out of 14 markers where observed heterozygosity was higher than expected [29]. The differences may be due to the use of different marker sets, or to differences in sample numbers.

The population genetic indices of subpopulation 2 are the same as the whole populations’ results, probably because 69.47% of the individuals belongs here.

No significant correlation between genetic and geographic distances was found, probably because of the high motility and reproduction rate of the wild boar. The lack of different populations in Hungary is consistent (i) with our previous results, where we focused on the effects of Hungary’s busiest highway [33] and (ii) with other researches’ results who found that although highways can cause increased mortality and decreased presence due to disturbance, they do not significantly affect the movement of wild boars [27,34,35,36]. Furthermore, natural barriers like high mountains, which could separate the populations [30,31] are also absent from the Carpathian Basin. The homogenous genetic structure was also confirmed in Vernesi’s study based on 29 wild boars [28]. The current situation could be explained by the effects of the last glacial and recent water management as well as the distribution and genetic diversity of wild boars, as in the case of most temperate European mammals, has been shaped by multiple climatic fluctuations [4,6,13,16,17], and also by human activities in the past centuries [9,28]. The last glaciation was followed by a sudden demographic and spatial expansion of wild boar populations [15,16], and until recently, human-induced gene flow (i.e., translocations, hybridisation) appeared to have had minor influence on the species [17,20,27].

## 5. Conclusions

Genetic characterization is essential for recognizing species diversity as well as for rationalizing managing activities. By the analysis of 13 STR markers of 486 wild boars, the genetic diversity is shown to be similarly high as found all across Europe. Although the marker set used was originally designed for Asian wild boars, with some optimizations it can also be used perfectly for European boars as a faster and cheaper alternative to previously used STR marker sets. In Hungary two subpopulations were found, which were almost completely separated by regions. The separation seems to be related only with the characteristics of the species and not with the recent geographical features, but traces of the period before water regulation seem to be visible. The genetical parameters of the subpopulations (and the whole Carpathian population) are diverse, but the high number of deviations from the Hardy-Weinberg equilibrium may cause greater differences between the subpopulations. The clarification of the exact location of the southern border between subpopulations necessitates the examination of more samples from Croatia, Serbia and Romania. In addition, our study has direct relevance for hunting strategies in wild boar. It is clear that, rather than relying on a general/national management strategy, subpopulations should be managed in an integrative way, strengthened with biological data.

## Figures and Tables

**Figure 1 genes-11-01194-f001:**
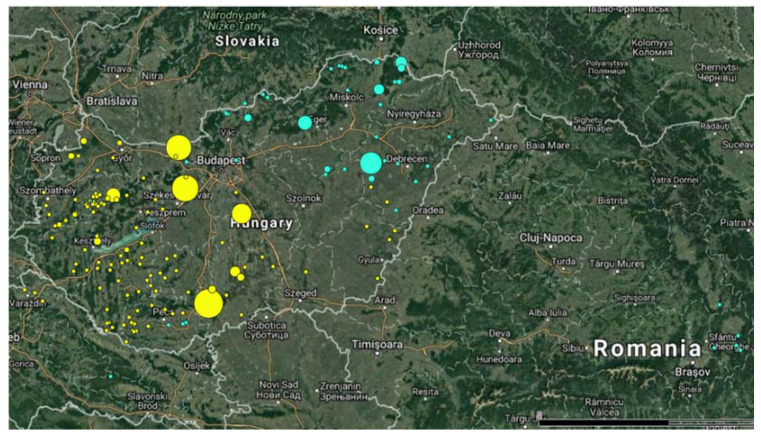
The geographic origin and genetic clustering of wild boar samples. The size of the circles represents the number of samples collected from a single hunting area. Blue dots: group 1 (*n* = 147), yellow dots: group 2 (*n* = 339).

**Figure 2 genes-11-01194-f002:**
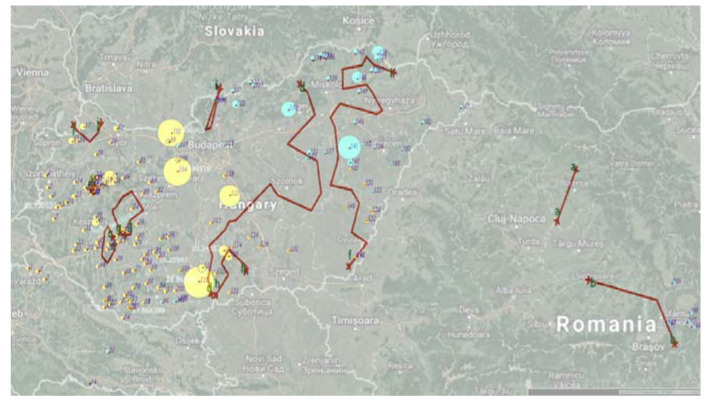
The results of the genetic analysis (Geneland) compared with the geographic analysis (Barrier). Red lines indicates the possible barriers to gene flow

**Figure 3 genes-11-01194-f003:**
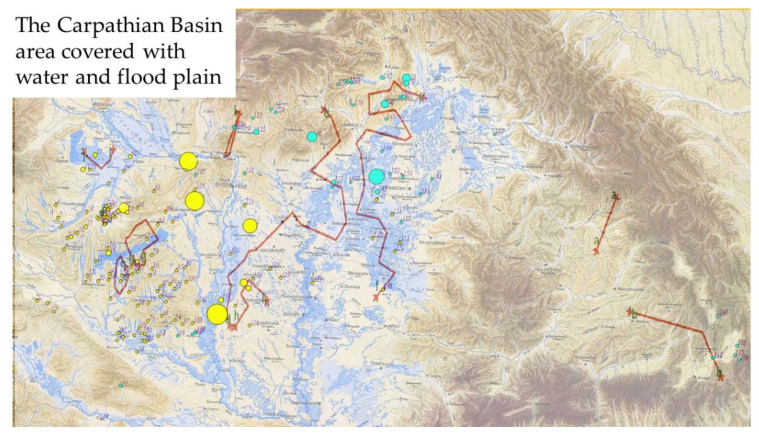
The results of the genetic analysis (Geneland) compared with the geographic analysis (Barrier), represented on the hydrographic map of the Carpathian Basin in the mid-1800′s. Red lines indicates the possible barriers to gene flow.

**Table 1 genes-11-01194-t001:** Genetic diversity found in the Hungarian wild boars studied.

Locus	N	Na	Ne	Ho	He	HWE
PigSTR14B	482	7	1.162	0.145	0.140	*
PigSTR7B	485	8	3.179	0.367	0.685	***
PigSTR4B	462	5	1.929	0.459	0.482	ns
PigSTR4C	485	6	1.891	0.404	0.471	***
PigSTR17A	485	6	1.450	0.353	0.311	ns
PigSTR11A	485	5	1.766	0.427	0.434	ns
PigSTR14A	477	4	1.640	0.375	0.390	ns
PigSTR11B	415	14	2.640	0.342	0.621	***
PigSTR1B	386	9	1.192	0.036	0.161	***
PigSTR15A	479	14	5.095	0.758	0.804	***
PigSTR5C	484	8	2.871	0.610	0.652	***
PigSTR13E	409	8	3.689	0.465	0.729	***
PigSTR1A	485	5	1.284	0.206	0.221	***

N: number of individuals genotyped, Na: number of alleles per loci, Ne: number of effective alleles per loci, Ho: observed heterozygosity, He: expected heterozygosity, HWE: deviation from the Hardy-Weinberg equilibrium (ns: not significant, *: *p* < 0.05, ***: *p* < 0.001).

**Table 2 genes-11-01194-t002:** Genetic diversity of Hungarian wild boars according to the clustering results.

Locus	Group 1 (*n* = 147)					Group 2 (*n* = 339)				
	Na	Ne	Ho	He	HWE	Na	Ne	Ho	He	HWE
PigSTR14B	4	1.11	0.10	0.10	ns	6	1.19	0.16	0.16	*
PigSTR7B	7	3.30	0.50	0.70	***	8	3.09	0.31	0.68	***
PigSTR4B	3	1.90	0.42	0.47	ns	5	1.94	0.48	0.48	ns
PigSTR4C	5	2.26	0.56	0.56	ns	6	1.75	0.34	0.43	***
PigSTR17A	6	1.66	0.49	0.40	ns	5	1.37	0.29	0.27	ns
PigSTR11A	4	2.08	0.61	0.52	*	4	1.58	0.35	0.37	ns
PigSTR14A	3	1.60	0.40	0.37	ns	3	1.66	0.37	0.40	ns
PigSTR11B	7	1.51	0.31	0.34	ns	13	3.08	0.35	0.68	***
PigSTR1B	5	1.10	0.04	0.09	***	6	1.23	0.03	0.19	***
PigSTR15A	12	4.84	0.75	0.79	ns	13	5.15	0.76	0.81	***
PigSTR5C	5	2.89	0.58	0.65	***	6	2.85	0.62	0.65	***
PigSTR13E	8	3.87	0.49	0.74	***	7	3.11	0.45	0.68	***
PigSTR1A	4	1.26	0.19	0.21	ns	3	1.29	0.21	0.23	***

Na: number of alleles per loci, Ne: number of effective alleles per loci, Ho: observed heterozygosity, He: expected heterozygosity, HWE: deviation from the Hardy-Weinberg equilibrium (ns: not significant, *: *p* < 0.05, ***: *p* < 0.001).

**Table 3 genes-11-01194-t003:** Review table of recent studies sorted by average number of alleles.

Sampling Site	No. of Individuals	No. of Markers	Range of Alleles	Avg. No. of Alleles	Reference
East Asia	238	16	-	3.4–9.6	[33]
Hungary	49	14	3–14	6.21	[29]
Germany	63	10	3–17	7.5	[30]
Hungary	486	13	4–14	7.62	Recent study
Belgium	325	14	5–25	8.8	[27]
Hungary	29	9	6–12	8.8	[28]
Croatia	264	14	4–19	8.92	[31]
Portugal	110	6	3–15	10.17	[32]
Bulgaria	289	10	5–31	12	[30]
Europe	723	11	9–29	19	[3]

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
