# Peer review of "Population Genetic Structure of the Wild Boar (Sus scrofa) in the Carpathian Basin"

_genes, 2020, doi:10.3390/genes11101194_

Round 1
Reviewer 1 Report
Overall, the manuscript brings useful information and closes a knowledge-gap.
On several occasions, you mention about conservation of the species (e.g. line 67 in the introduction). I believe this is hardly the case, as in most countries it is regarded as a pest and its population is regulated through hunting.
The methodology is simple and well described.
My main point of concern is directed towards the discussion section. There is no mentioning of presence or lack of morphological differences among the two major genetic subpopulations. Furthermore, what are the hypotheses for this population genetic situation?
And a full paragraph on the evolution in space and time of this species across Hungary and Europe is needed. When and from where did the wild boars reached Europe? How did the last glaciations influence its dispersal and current distribution range?
My feeling is that the reader will not understand why in present we have this genetic structure of wild boars in the Carpathian Basin and in Europe. I encourage you to tackle this gap in your manuscript.
Author Response
Dear Reviewer,
First of all, we would like to thank you for your work and your comments, making it possible for us to improve the quality of our article. Please find our answers below.
Q: On several occasions, you mention about conservation of the species (e.g. line 67 in the introduction). I believe this is hardly the case, as in most countries it is regarded as a pest and its population is regulated through hunting.
A: We totally agree with that, and in the manuscript we wrote “in extreme cases”, by which we meant the unforeseen effects of ASF, for example. Nevertheless, it is quite unlikely that wild boar will ever become an endangered species, so these parts have been deleted.
Q: My main point of concern is directed towards the discussion section. There is no mentioning of presence or lack of morphological differences among the two major genetic subpopulations. Furthermore, what are the hypotheses for this population genetic situation? And a full paragraph on the evolution in space and time of this species across Hungary and Europe is needed. When and from where did the wild boars reached Europe? How did the last glaciations influence its dispersal and current distribution range? My feeling is that the reader will not understand why in present we have this genetic structure of wild boars in the Carpathian Basin and in Europe. I encourage you to tackle this gap in your manuscript.
A: In this study no data were measured from which phenotypic segregation could be established. From our own findings and from the lack of literature data we conclude that there are no well-defined phenotypical differences between subpopulations. In our opinion, the current situation is partly the result of the last ice age and mainly preserves traces of the period before the Hungarian water regulation, which took place in the late 1800s. A new paragraph and two figures were added to the Introduction and the Discussion on this subject.
We hope you find our responses and amendments acceptable, and our collaboration will be a success,
Yours faithfully,
Bendegúz Mihalik

Reviewer 2 Report
Mihalik et al. in their article described genetic diversity of wild boar in Hungary. They used 13 STR markers for 486 samples. The aim of the study is given in the introduction. This is not the first study on wild boar population from this region, many different papers with analyses of different molecular markers have been published previously. In this article, the dataset is impressive, however no novel conclusions were made and study area is quite narrow ('one country' scale).
This article is a simple description of genetic diversity of wild boars. In my opinion, there is no deeper explanation why this study is important. Authors only mentioned that different studies were done previously, however no comparison is presented in the Discussion. Is there any congruence between results and genetic patterns obtained from mtDNA and STR? Why there are 2 groups in Hungary if there is no geographic barriers? Is there any barrier to gene flow? I would recommend BAPS which allow to analyze genetic structure in reference to spatial data and BARRIER to check where the barriers are.
The ASF is now important issue in Europe, which significantly shape genetic structure of wild boar. Is ASF problem in Hungary? Is ASF had any influence on observed genetic structure in Hungary?
Therefore, I am not able to recommend this version of article for publishing.
Author Response
Dear Reviewer,
First of all, we would like to thank you for your work and your comments, making it possible for us to improve the quality of our article. Please find our answers below.
Q: Mihalik et al. in their article described genetic diversity of wild boar in Hungary. They used 13 STR markers for 486 samples. The aim of the study is given in the introduction. This is not the first study on wild boar population from this region, many different papers with analyses of different molecular markers have been published previously. In this article, the dataset is impressive, however no novel conclusions were made and study area is quite narrow ('one country' scale). This article is a simple description of genetic diversity of wild boars. In my opinion, there is no deeper explanation why this study is important.
A: Although our study is not the first within the Carpathian Basin, we analyzed a much larger number of samples than previous studies. Also, Hungary is located in the middle of the Carpathian Basin, which represents a crossroads of recolonization routes and a biodiversity hotspot for many terrestrial species. However, the genetic and population structures of species in this region is not well documented, therefore any new information from this area is important. A new section has been added to the Introduction, the Results and the Discussion to better emphasize the importance of our research.
Q: Authors only mentioned that different studies were done previously, however no comparison is presented in the Discussion.
A: We improved the manuscript by illustrating the comparison with a new table in the Discussion section.
Q: Is there any congruence between results and genetic patterns obtained from mtDNA and STR?
A: In this study we didn’t look at patterns obtained from mtDNA, in general, the results of the two methods are less comparable than those of different studies using STR markers.
Q: Why there are 2 groups in Hungary if there is no geographic barriers?
A: In our opinion, the current situation is partly the result of the last ice age and mainly preserves traces of the period before the water regulation of the Danube and the Tisza and the drainage of fields, which took place in the late 1800s. A new map and some information was added to the Discussion on this subject.
Q: Is there any barrier to gene flow? I would recommend BAPS which allow to analyze genetic structure in reference to spatial data and BARRIER to check where the barriers are.
A: In our study we used Geneland software, which is a similar and slightly newer program than BAPS. Using this software we analyzed the genetic structure in reference to geographical data, which is shown in Figure 1 in the manuscript. In addition, following your suggestion we performed an analysis with the BARRIER software; the results can be found in the Results.
Q: The ASF is now important issue in Europe, which significantly shape genetic structure of wild boar. Is ASF problem in Hungary? Is ASF had any influence on observed genetic structure in Hungary?
A: ASF has been officially present in Hungary since the spring of 2018, although, according to unofficial verbal statements, infected individuals had also been found a few months earlier. Based on the number of individuals hunted, the decrease is significant: in 2017 there was an increasing tendency with 158,079 wild boars hunted, in contrast to 148,265 in 2018 and 128,545 in 2019. Most of our sampling took place before 2018, so unfortunately we could not study its effect on genetic diversity. Nonetheless, this issue is really interesting and important, which would require further sampling to clarify.
We hope you find our responses and amendments acceptable, and our collaboration will be a success,
Yours faithfully,
Bendegúz Mihalik

Round 2
Reviewer 1 Report
Congratulations for the improvements. The paper looks much better in the current version.
Goodluck with your research!
Reviewer 2 Report
Authors addressed all my concerns.